# Polymers Containing Non-Covalently Bound Cyclodextrins

**DOI:** 10.3390/polym11030425

**Published:** 2019-03-05

**Authors:** Alan E. Tonelli

**Affiliations:** North Carolina State University, Wilson College of Textiles, Fiber & Polymer Science Program, Campus Box 8301 1020 Main Campus Drive, Raleigh, NC 27606-8301, USA; atonelli@ncsu.edu; Tel.: +1-191-515-6588

**Keywords:** polymers, cyclodextrins, non-stoichiometric inclusion complexes

## Abstract

We summarize and review the formation, characterization, behaviors, and possible uses of polymers that are threaded through, but only partially covered by cyclodextrins (CDs), which we call non-stoichiometric polymer–CD inclusion compounds (ICs) or non-stoichiometric (n-s) polymer–CD ICs. Emphasis is placed on comparison of the behaviors of unthreaded neat polymers with those that are threaded through and partially covered by CDs. These comparisons lead to several suggested uses for (n-s) polymer–CD ICs.

## 1. Introduction

When an excess of guest polymers is used to form inclusion compounds with host cyclodextrins (CDs), non-stoichiometric polymer–CD inclusion compounds (ICs) [(n-s) polymer–CD ICs] can result [1,2,3,4,5] and are depicted schematically in Figure 1. The unthreaded polymer chain portions in (n-s) polymer–CD ICs are then free to associate with each other and potentially crystalize. At the same time, the threaded chain portions covered with host CDs can crystallize in the same crystal structures assumed by stoichiometric polymer–CD ICs, with fully covered threaded guest polymer chains.

This results in shape-memory behavior [6]. When the unthreaded chain portions are heated above their *T*_g_s and *T*_m_s, the (n-s) polymer–CD ICs may be deformed and then cooled below their *T*_g_s and *T*_m_s and solidified to retain their deformed shapes. Then, their original shapes are quickly recovered after heating the unconstrained deformed samples above their *T*_g_s and *T*_m_s.

In addition, it has been observed that the unthreaded chain portions of (n-s) polymer–CD ICs are more readily crystallizable or have higher glass-transition temperatures than their neat samples [1,2,3,4,5,6,7,8,9,10]. This is demonstrated in Figure 2, where the differential scanning calorimetric (DSC) cooling curves of neat nylon-6 and the 3:1 (nylon-6:α–CD) (n-s) α–CD IC are presented. Note that the unthreaded molten nylon-6 chain portions in the (n-s) nylon–6–α–CD IC crystallize ~35 °C higher than the neat molten nylon-6 chains. As a consequence, (n-s) polymer–CD ICs may be added in small amounts to their neat polymer samples and, as shown in Figure 3 for nylon-6, can nucleate the crystallization of the neat unthreaded polymer chains [6,8].

We will now review and summarize the formation and behaviors of (n-s) polymer–CD ICs, using nylon-6 as an example. However, many different polymers have been observed to form stoichiometric and (n-s) CD ICs, as described only selectively in references [1,2,3,4,5,6,7,8,9,10].

## 2. Formation of (n-s) Polymer–CD ICs

A 3:1 (nylon-6:α–CD) (n-s) IC [8] was formed by dissolving at room temperature 1.5 g of nylon-6 (MW = 60,000) in 45 m of 90% formic acid. To this solution, 180 mL of 99% acetic acid was added. Then, 4.27 g of α-CD was dissolved in 21 mL of 99% dimethyl-sulfoxide and added to the nylon-6 solution. This was followed by stirring for 2 h at 50 °C and 6 h at room temperature. The formed precipitate was vacuum-filtered and dried in a vacuum oven.

Films of nylon-6 containing small amounts of (n-s) nylon-6–α–CD ICs and, for comparison, Talc as nucleants, were produced by dissolving nylon-6 in 90% formic acid, suspending the nucleants and stirring, and evaporating the formic acid.

## 3. Characterization of (n-s) Polymer–CD ICs

### 3.1. FTIR

Infrared observations were conducted with a Nicolet 510P Fourier transform infrared (FTIR) spectrometer in the range of 4000 to 400 cm^−1^, with a resolution of 2 cm^−1^. For the FTIR absorption measurements, powdered samples of nylon-6 and (n-s) nylon-6–α–CD IC were pressed into KBr pellets.

### 3.2. DSC

A Perkin Elmer Diamond DSC-7 instrument was used to record differential scanning calorimetric (DSC) thermal scans over the range of 25 °C to 250 °C at heating and cooling rates of 10 °C/min, 50 °C/min, 100 °C/min, 150 °C/min, and 200 °C/min. Nitrogen was used as the purge gas.

### 3.3. WAXD

Powdered samples were observed by wide-angle X-ray diffraction (WAXD) with a Siemens type-F X-ray diffractometer. A Ni-filtered Cu K_α_ radiation source (λ = 1.54 A°) was employed, and the supplied voltage and current were 30 kV and 20 mA, respectively. Diffraction intensities were measured every 0.1° from 2θ = 5 to 30° at a rate of (2θ = 3°)/min.

### 3.4. Microscopy

A Nikon Eclipse 50i POL polarizing optical microscope was used to record micrographs of melt crystallized N-6 films. Images were captured on a Sony Corp. CCID-IRIS/RGB color video camera.

### 3.5. Shape-Memory Testing

Melt-pressed films of neat nylon-6 (60,000 g/mol) and its 3:1 (n-s) α–CD IC (two samples each) were suspended over closely adjacent and inverted petri dishes. On top of each film, which were ~6.5 cm in length, ~0.8 cm in width, and ~0.15 mm thick, a 2-g weight was placed on top of the suspended portion. They were placed in a vacuum oven and heated to 250°C. The small weight was removed after cooling to room temperature, and the films were then returned to the 250 °C oven [6].

## 4. Characterization Results

### 4.1. FTIR

In Figure 4, vibrational bands from both guest nylon-6 and host α–CD are present in the FTIR spectra of stoichiometric (1:1) and (n-s) (3:1) nylon-6–α–CD ICs. The latter sample not surprisingly contains an expectedly greater proportion of vibrational bands from the nylon-6 guest.

### 4.2. DSC

From Figure 3, it is apparent that the unthreaded chains in the neat (n-s) nylon-6–α–CD IC and the nylon-6 sample containing 2 wt % of the (n-s) nylon-6–α–CD IC crystallize more readily (narrower crystallization exotherms) at higher temperatures as they are cooled from their melts in comparison to the molten neat nylon-6 sample.

### 4.3. WAXD

In Figure 5c, diffraction peaks attributable to both the columnar structure [11] of the host α–CD crystalline lattice (as in Figure 5b) and those portions of the nylon-6 chains “dangling” from them that had crystallized (as in Figure 5d) are visible in the WAXD of the 3:1 (n-s) nylon-6–α–CD IC.) The diffractograms of as-received nylon-6 and the nylon-6 coalesced from its stoichiometric α–CD IC upon treatment with dilute HCl in part (d) show, respectively, the presence of both the α-nylon-6 and γ-nylon-6 polymorphs, and predominantly the α-polymorph [8]. The unincluded “dangling” nylon-6 chains in the 3:1 (n-s) nylon-6–α–CD IC are also crystallizing predominantly in the α-polymorph from the comparison of (b), (c), and (d). Since in the α-polymorph neighboring chains are antiparallel, the nylon-6 chains in neighboring α–CD IC channels may also be included in antiparallel directions. This would imply that the nylon-6 chains “dangling” from the same α–CD IC crystal are folding and may be reentering the same crystal, but in adjacent α–CD IC channels, and are included in and contributed by different α–CD IC crystals. This latter suggestion is supported by the shape-memory behavior exhibited by (n-s)–polymer–CD ICs [6].

### 4.4. Microscopy

Figure 6 clearly shows that the addition of small amounts of (n-s) nylon-6–α–CD ICs to neat bulk nylon-6 results in melt-crystallized semicrystalline morphologies that are both more uniform and finer in scale than those in melt-crystallized neat nylon-6. Though not shown here, we have observed with DSC that both the crystallization temperatures and levels of crystallinity achieved from the non-isothermal crystallization of nylon-6 samples containing (n-s) nylon-6–α–CD ICs as potential nucleants are higher than for neat nylon-6 samples [6]. Thus, (n-s) nylon-6–CD ICs appear to be very effective nucleants for nylon-6, providing a non-toxic alternative to commonly used nucleants such as talc [6], mica, or clays.

### 4.5. Shape-Memory Testing

When heated, the neat 3:1 (n-s) N-6–α–CD IC films softened and sagged very slightly between the supporting petri dishes, but remained integral without flowing, while they sagged dramatically when carrying the 2-g weight. Upon heating, the neat nylon-6 film melted completely and flowed down the sides of both supporting petri dishes. The severely sagging 3:1 (n-s) nylon-6–α–CD IC film, upon removal of the small weight and returning it to the 250 °C oven, retracted almost completely. It appeared closely similar to the 3:1 (n-s) nylon-6–α–CD IC film without the weight that was heated to 250 °C [8]. These observations, though only a qualitative test for shape-memory behavior, strongly suggest that the CD IC crystalline domains containing included portions of the nylon-6 chains act as cross-links in (n-s) nylon-6–α–CD ICs, which are essentially networks. When heated above T_m_, nylon-6 chains and the (n-s) nylon-6–α–CD IC network as a whole may be stretched. In addition, this implies that the unincluded portions of at least some nylon-6 chains do not actually “dangle”, but instead are constrained on both ends by inclusion in different CD IC crystals.

### 4.6. Summary of Characterization Results

When excess nylon-6 is employed, non-covalently bonded crystalline non-stoichiometric (n-s) nylon-6–CD ICs were formed by threading the host cyclic CD starches onto guest nylon-6 chains.

The constrained crystallization of nylon-6 chains protruding from the (n-s) nylon-6–CD ICs has been compared to the crystallization of bulk nylon-6 samples and found to be faster and occurring at higher temperatures.

Though not shown here, the protruding, unthreaded, and constrained chains in (n-s) nylon-6–α–CD IC “brushes” crystallize faster and to a significantly greater extent than the nylon-6 chains dangling from the surfaces of (n-s) nylon-6–γ–CD ICs. Although the unincluded nylon-6 chains protruding from the surfaces of (n-s) γ–CD ICs also constitute high-density “brushes” [12], whose chains are highly extended and aligned perpendicular to the surfaces of their attachment (see Figure 7), strong interactions between the pairs of nylon-6 chains emerging from each γ–CD IC channel may reduce the ability of such a resulting nylon-6 dense “brush” to crystallize. When added at low concentrations, the non-toxic, biodegradable/bioabsorbable (n-s) nylon-6–α–CD ICs and (n-s) nylon-6–γ–CD ICs both serve as effective nucleating agents for the melt crystallization of bulk nylon-6.

As indicated in Figure 7, the unincluded nylon-6 chains “dangling” from their (n-s) α–CD ICs, as they emerge from ~0.5-nm channels, are separated by ~1.4 nm compared to a pair of closely adjacent nylon-6 chains in each ~1 nm γ–CD IC channel, separated by ~1.7 nm. The differences in the constraints imposed on their unincluded chains “dangling” from their CD IC channels may be, at least in part, the source for the distinct behaviors observed for (n-s) nylon-6–CD ICs made with α–CDs and γ–CDs (more extensive discussion of this topic can be found in [13]).

## 5. Uses for (n-s) Polymer–CD ICs

### 5.1. As Melt-Crystallization Nucleants

We have already demonstrated in Figure 2 and Figure 3 that the melt crystallization of nylon-6 can be nucleated and accelerated through the addition of small amounts of (n-s) nylon-6–α–CD IC. In Figure 8, we show the effects of adding small quantities of (n-s) nylon-6–CD ICs and talc to neat nylon-6 and follow their crystallization from the melt. Even though nylon-6 in both neat and 6:1 (n-s) γ–CD IC samples, as mentioned previously, seem to crystallize similarly, note that the 6:1 (n-s) nylon-6–γ–CD IC nevertheless seems to effectively nucleate the melt crystallization of bulk nylon-6. In fact, judging from the DSC cooling scans in Figure 9, 6:1 (n-s) nylon-6–γ–CD ICs appear to be more effective nucleants than the 3:1 (n-s) nylon-6–α–CD ICs. There, it is also seen that the 6:1 (n-s) nylon-6–γ–CD IC is as effective a nucleant as talc.

Even though the unincluded portions of nylon-6 chains in the 6:1 nylon-6–γ–CD IC were observed to crystallize to a lesser extent and at a lower temperature than those in the 3:1 (n-s) α–CD IC [8], they were more effective at nucleating bulk nylon-6. The presence of side-by-side pairs of nylon-6 chains “dangling” from the surfaces of (n-s) nylon-6–γ–CD IC crystals may explain this behavior. The morphology produced by the 3:1 (n-s) nylon-6–α–CD IC in Figure 6, on the other hand, appears more homogeneous and of a finer grain than that seen, but not shown, in the nylon-6 sample nucleated with 6:1 (n-s) nylon-6–γ–CD IC. Without determining the particle sizes of the nucleants used here, the reasons that have been offered to explain observed differences in the melt crystallized morphologies produced by them remain tentative [2,3,4].

In Figure 9, the ability of nylon-6 coalesced from its 3:1 (nylon-6:α–CD) (n-s α–CD IC) to nucleate the melt crystallization of bulk neat nylon-6 is compared to the nucleation ability of nylon-6 chains coalesced from its completely threaded and covered 1:1 nylon-6–α–CD IC.

We see that both show nearly identical abilities to enhance the melt crystallization of neat bulk nylon-6. A comparison of Figure 8 and Figure 9 shows that the nylon-6 chains coalesced from their α–CD ICs are more effective nucleants than the unthreaded nylon-6 chains in their (n-s) α–CD ICs. The coalesced nylon-6 samples contain no α–CD, and so we characterize them as “stealth” nucleants, because they introduce nothing but nylon-6 chains to the neat bulk nylon-6 samples they nucleate.

In addition, once a melt-crystallized neat bulk polymer has been nucleated with the same polymer coalesced from its CD ICs, the nucleated sample can also be used as a nucleant to enhance the melt crystallization of additional neat bulk polymer samples. This is demonstrated in Figure 10 for poly (ε-caprolactone) (PCL), where after an initial nucleation with coalesced PCL two fresh PCL samples are nucleated with the previously nucleated PCL samples. The net result is that only 0.001% of the initial coalesced PCL nucleant is present in the final sample of nucleated PCL.

### 5.2. As Potential Compatiblizers

#### CD-Stars

A star polymer with a γ–CD core and polystyrene (PS) arms (CD-star) (see Figure 11) was used to partially compatibilize blends of the highly immiscible polymers polystyrene (PS) and poly(dimethyl-siloxane) (PDMS), even though the latter is a liquid [14]. Threading of the γ–CD core by PDMS and the subsequent solubilization in the PS matrix is facilitated by the PS star arms and likely constitutes the mechanism of compatibilization.

Films cast from clear solutions in chloroform exhibit large wispy PDMS domains, indicating that some dethreading of the γ–CD-star and agglomeration of PDMS takes place during the slow process of solvent evaporation. However, DSC and dynamic mechanical analysis (DMA) data show that partial compatibilization takes place, as evidenced by a shift in the *T*_g_s of PS and PDMS toward each other. Compared to samples without γ–CD-stars, the shift in the PS *T*_g_ is greater when the γ–CD-star is present. During solvent evaporation and the post-processing of the films, PDMS tends to leach out of solution-cast films. However, the amount of retained PDMS is significantly increased when the γ–CD-star is present, as is evidenced in Figure 12. As little as a 1.0 wt % γ–CD core concentration is observed to compatibilize the PDMS–PS blend. Though not presented here, DMA data also show that the γ–CD-stars reduce the molecular mobility of the threaded PDMS chains [14].

Micrographs of spun-cast films containing varying amounts of PDMS with and without a 1 wt % CD core are shown in Figure 13. There, it is very clear that the γ–CD stars with PS arms serve to compatibilize the PS/PDMS blends containing significant quantities of the normally neat liquid PDMS. As seen in Figure 14, these compatibilized spun-cast PS/PDMS blends remain well mixed even after heating them under N_2_ well above the *T*_g_ of PS at 125 °C for 3 days. Even though PS and PDMS chains should be mobile at this temperature, after annealing for 3 days, phase separation was only seen to occur for the spun-cast film containing 20 wt % PDMS.

## 6. Viscosity Control of Polymer Solutions with CDs

Associative polymers are macromolecules with attractive hydrophobic groups either attached to the chain ends or randomly distributed along the backbone. The solution rheology of a comblike, hydrophobically modified alkali-soluble emulsion (HASE) associative polymer shown in Figure 15 was modified through addition of α-CDs and *β*-CDs [15,16]. The interactions of the hydrophobic inner cores with the pendant macromonomer segments of the associative polymer containing hydrophobic end groups were disrupted by the CD threading, and led to the dramatic reduction in polymer solution viscosity (see Figure 16).

The CDs encapsulated the hydrophobic groups on the associative polymer, as confirmed by ^1^H NMR, DSC, and TGA observations of the complexation between the CDs and a surfactant that was modified to resemble the hydrophobic macromonomer of the associative polymer. The stoichiometric ratio of complexation between the CDs and the hydrophobic macromonomer was determined to be 5 mol of CD/mol of hydrophobe. Interestingly and importantly, the reduction in polymer viscoelasticity in the presence of CD is reversibly recovered upon the subsequent addition of different nonionic surfactants that have a higher propensity to complex with the CD than the hydrophobic segments of the HASE polymer.

## 7. Summary

Above, we have briefly summarized the formation, characterization, and some uses for (n-s) polymer–CD ICs containing dangling unthreaded portions of partially included guest polymer chains, whose behaviors were compared to the neat polymers. The effects of the host CD cavity size and guest polymer chain occupancy (single or pairs of threaded chains) were also discussed. Most importantly, the ability of unthreaded chain portions of (n-s) polymer–CD ICs and the guest polymers coalesced from them to nucleate the melt crystallization of the same neat bulk polymer samples was illustrated. This leads to melt crystallized polymer materials with more homogeneous finer scale morphologies that lead to improved mechanical properties [4,17,18]. It also makes their use for more easily recyclable materials and implantable devices possible.

Star molecules formed by appending PS arms onto γ–CD cores were shown to be effective in compatibilizing normally highly incompatible PS/PDMS blends [14,15]. These blends were seen to remain well-mixed solids, even after long-time annealing above the *T*_g_ of PS. Compared to conventional block copolymer compatibilizers, this novel type of compatibilizer could be advantageous, since many different polymers are capable of threading CD cores. The same CD star molecule could be used to compatibilize several different A/B polymer blends in which polymer B is varied. Since α–CD, β–CD, and γ–CD have different cavity diameters, as the core of CD-stars, this also allows the compatibilizer to be tailored for the selective threading of the desired polymers.

We also demonstrated that the solution rheology of a comblike, hydrophobically modified alkali-soluble emulsion associative polymer could be modified and controlled through the addition of CDs, which thread over the hydrophobic chain portions, prevent their association, and lead to dramatically reduced solution viscosities [16,17].

## Figures and Tables

**Figure 1 polymers-11-00425-f001:**
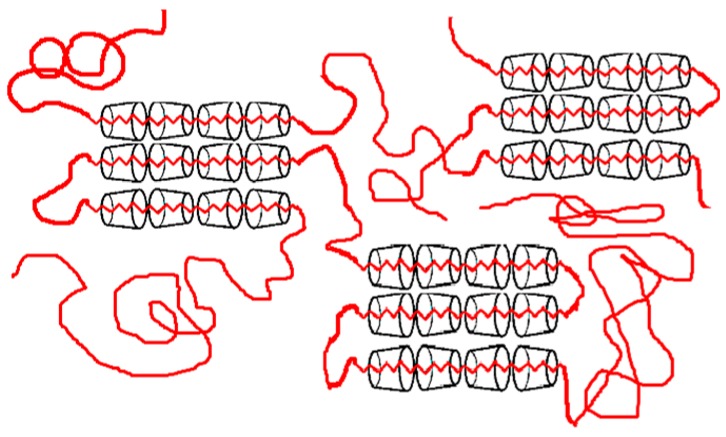
Schematic drawing of a non-stoichiometric (n-s) polymer–cyclodextrin (CD) inclusion compounds (IC).

**Figure 2 polymers-11-00425-f002:**
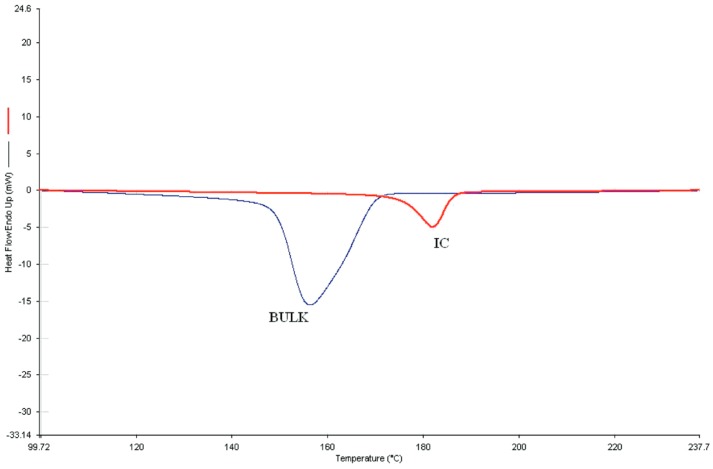
Differential scanning calorimetric (DSC) cooling scans (10 °C/min) from the melts of 600,000 g/mol bulk nylon-6 and its 3:1 (nylon-6:α–CD) (n-s) α–CD–IC, reproduced from [8] with permission, copyright American Chemical Society, 2009.

**Figure 3 polymers-11-00425-f003:**
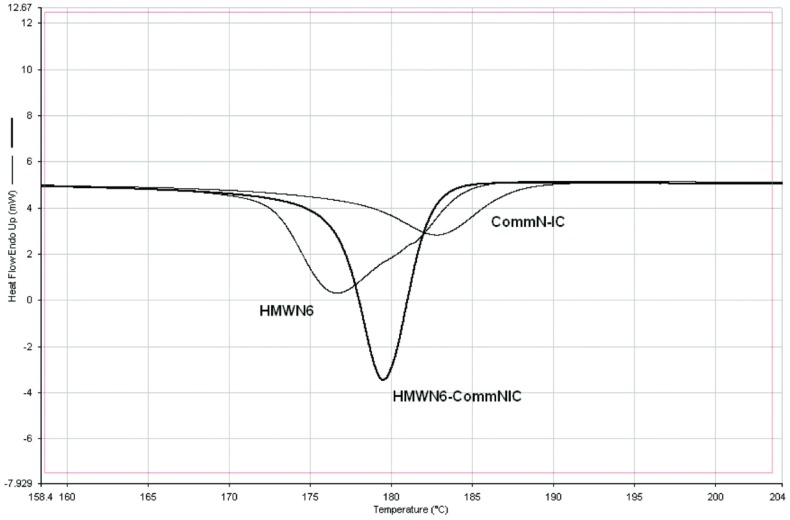
Crystallization on cooling (10 °C/min) melts of neat nylon-6 (HMWN6), 3:1 (n-s) nylon-6–α–CD IC (CommN-IC), and nylon-6 with 2 wt % of 3:1 (n-s) nylon-6–α–CD IC (HMWN6-CommNIC). In each case, nylon-6 with MW = 60,000 was employed, reproduced from [8] with permission, copyright American Chemical Society, 2009.

**Figure 4 polymers-11-00425-f004:**
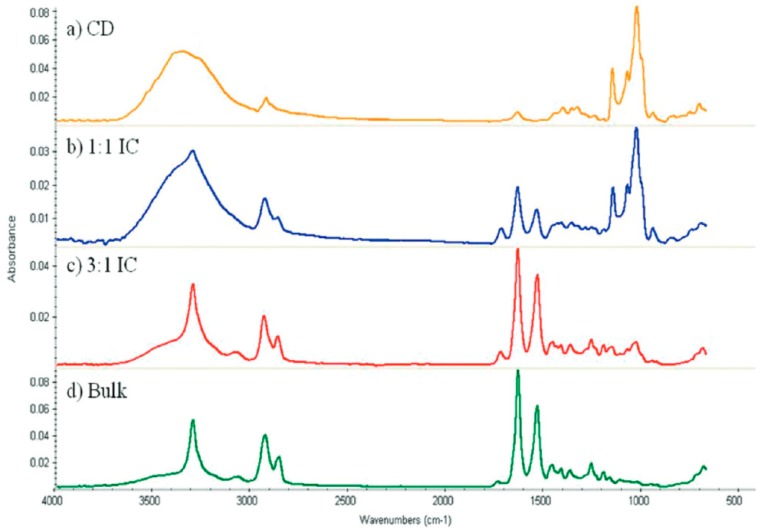
Fourier transform infrared (FTIR) spectra for (**a**) α–CD, (**d**) 600,000 g/mol nylon-6, and their (**b**) 1:1 stoichiometric and (**c**) 3:1 (n-s) α–CD ICs.

**Figure 5 polymers-11-00425-f005:**
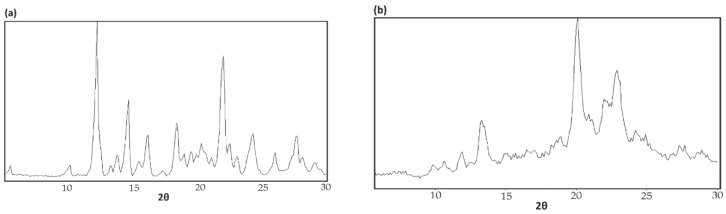
Wide-angle X-ray diffraction (WAXD) diffractograms for as-received cage α–CD (**a**), stoichiometric 1:1 nylon-6–α–CD IC (**b**), 3:1 (n-s) nylon-6–α–CD IC (**c**), and (**d**) as-received nylon-6 (left) and nylon-6 from its stoichiometric 1:1 nylon-6–α–CD IC (right). Reproduced from [8] with permission, copyright American Chemical Society, 2009.

**Figure 6 polymers-11-00425-f006:**
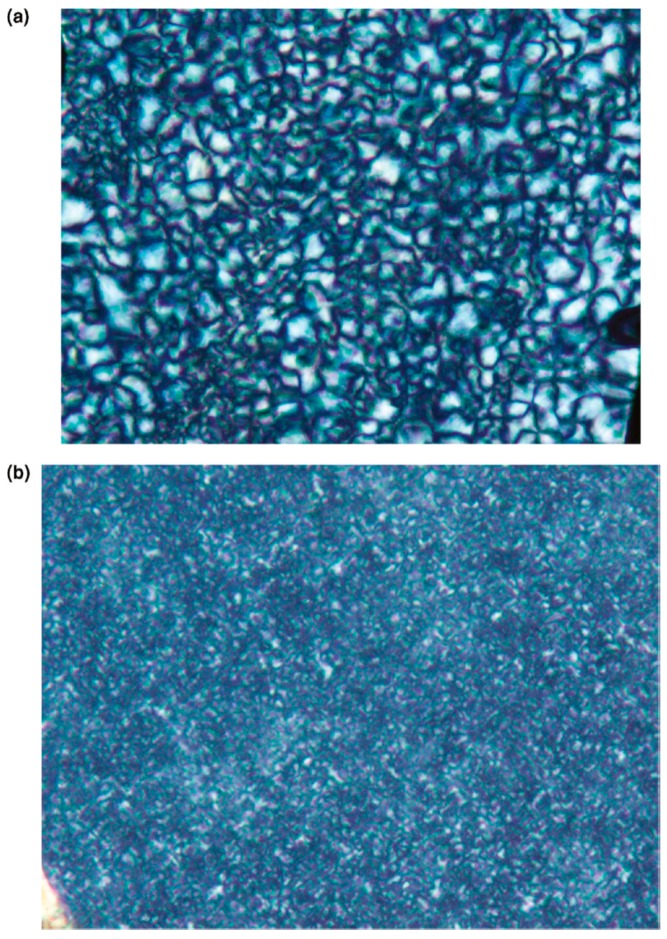
(**a**) Polarized micrograph of nylon-6 (MW = 600,000) film crystallized from the melt. (**b**) Polarized micrograph of nylon-6 (MW = 600,000) film with 2 wt % 3:1 (n-s) nylon-6 (MW = 600,000) α–CD IC crystallized from the melt, reproduced from [8] with permission, copyright American Chemical Society, 2009.

**Figure 7 polymers-11-00425-f007:**
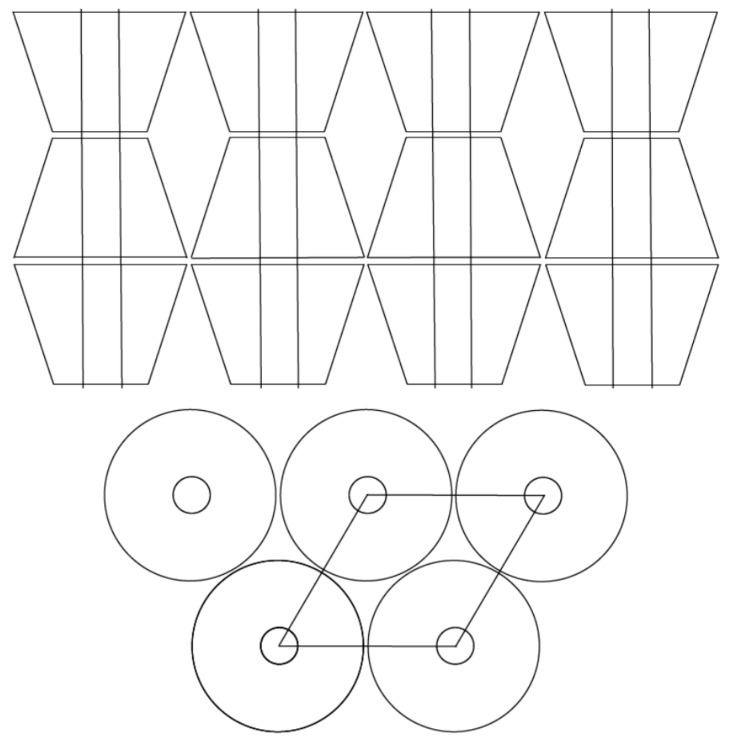
Channel structure of a columnar CD IC, with 0.6 (α–CD) or 0.8 (γ–CD) nylon-6 chains/nm^2^ of CD IC crystal surface. Threaded chains are ~1.5–1.8 nm apart, so the protruding nylon-6 chains form dense polymer brushes, [12] (for comparison, in the α-form bulk crystal, there are 1.4 nylon-6 chains/nm^2^) [8].

**Figure 8 polymers-11-00425-f008:**
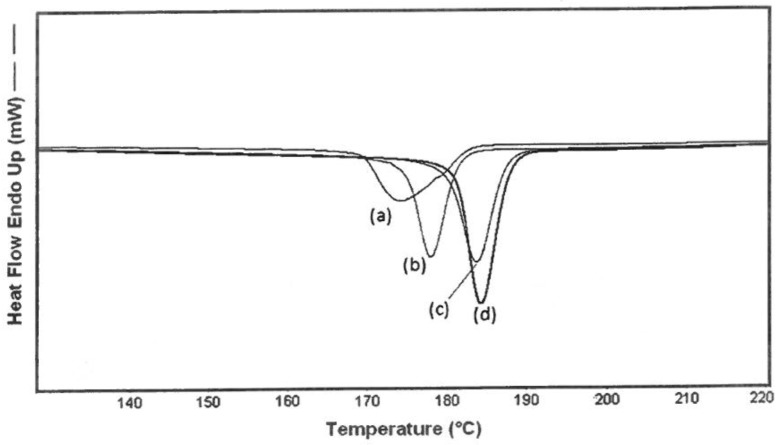
DSC cooling scans (5 °C/min) for neat (**a**) nylon-6 sample (600,000 g/mol) (**b**) nylon-6 with 2 wt % 3:1 (n-s) nylon-6–α–CD IC, (**c**) nylon-6 with 2 wt % talc, and (**d**) nylon-6 with 2 wt % 6:1 (n-s) nylon-6–γ–CD IC. Reproduced from [6] with permission, copyright Elsevier, 2009.

**Figure 9 polymers-11-00425-f009:**
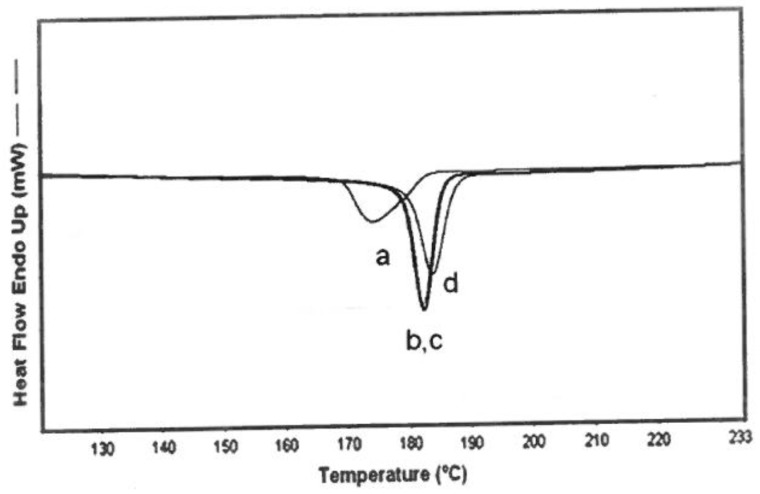
DSC cooling scans for (**a**) neat nylon-6 sample (600,000 g/mol), with 2 wt % of (**b**) nylon-6 coalesced from the 1:1 fully threaded nylon-6–α–CD IC, (**c**) nylon-6 coalesced from 3:1 (n-s) nylon-6–α–CD IC, and (**d**) nylon-6 with 2 wt % talc. Reproduced from [6] with permission, copyright Elsevier, 2009.

**Figure 10 polymers-11-00425-f010:**
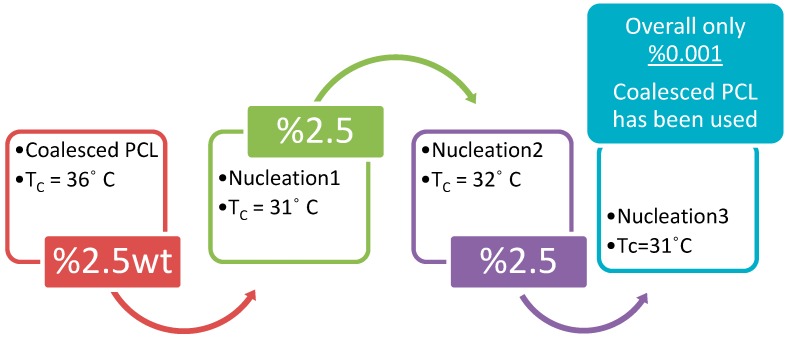
Repeated nucleation of the melt crystallization of poly (ε-caprolactone) (PCL) using coalesced PCL initially, followed by the repeated use of nucleated PCL to nucleate the melt crystallization of fresh bulk PCL. It should be noted that neat bulk PCL crystallizes at ~11 °C at the same DSC cooling rate of 20 °C/min.

**Figure 11 polymers-11-00425-f011:**
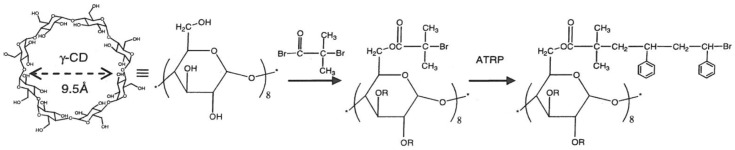
Synthesis of star polymers with a γ–CD core and polystyrene (PS) arms (CD-stars). Reproduced from [14] with permission, copyright Elsevier, 2010.

**Figure 12 polymers-11-00425-f012:**
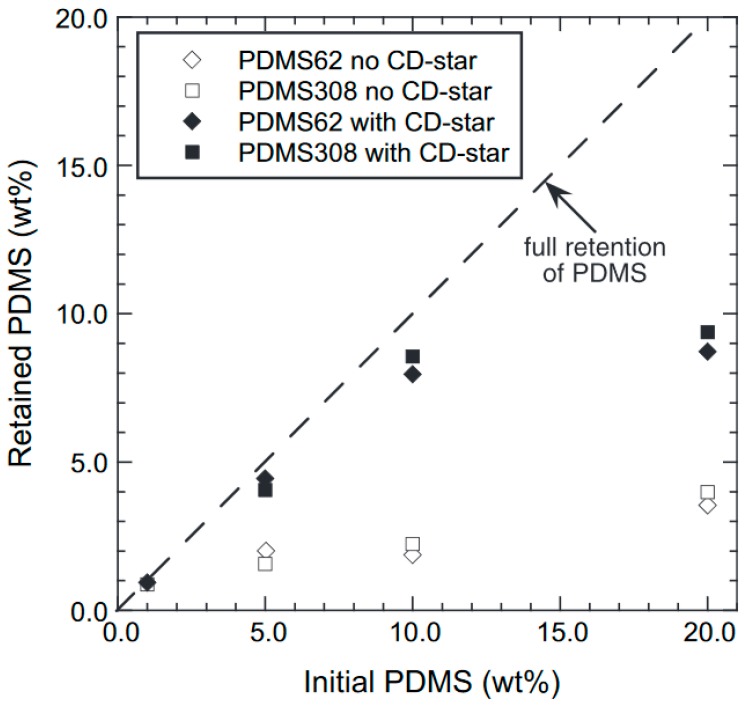
Retained poly(dimethyl-siloxane) (PDMS) (MWs = 62,000 and 308,000) vs. initial PDMS content for solution-cast films with and without γ–CD-stars. Films containing γ–CD-stars have a γ–CD core concentration of 1.0 wt %. Reproduced from [14] with permission, copyright Elsevier, 2010.

**Figure 13 polymers-11-00425-f013:**
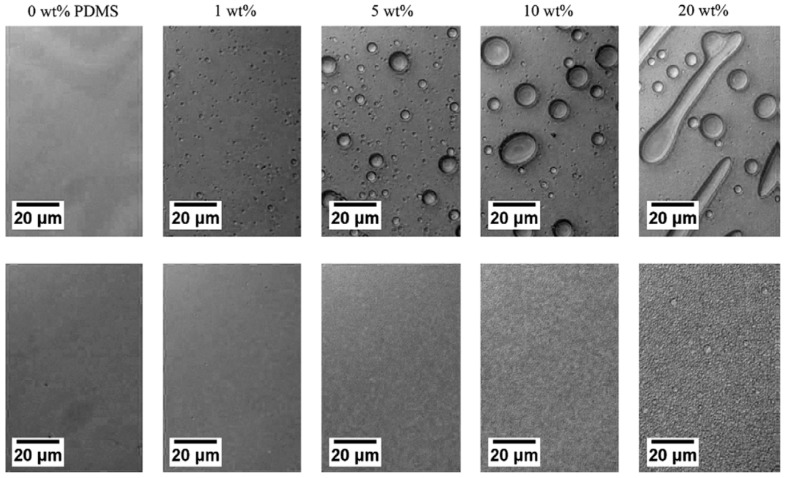
Reflection optical micrographs of PS/PDMS (MW = 62,000) spun-cast films with varying amounts of PDMS. Top row: blends without CD-star; Bottom row: blends containing CD-star at 1 wt % CD core. Reproduced from [14] with permission, copyright Elsevier, 2010.

**Figure 14 polymers-11-00425-f014:**
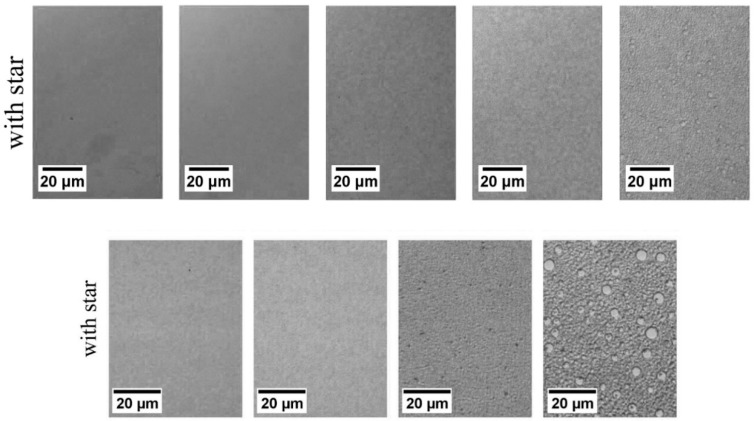
Compatibilized spun-cast films with constant 1 wt % CD core. The top row of films contain 0 wt %, 1 wt %, 5 wt %, 10 wt %, and 20 wt % PDMS (MW = 308,000) from left to right. The bottom row of films contain 1 wt %, 5 wt %, 10 wt %, and 20 wt % PDMS, and were thermally held above the *T*_g_ of PS at 125 °C for 3 days under N2. Reproduced from [14] with permission, copyright Elsevier, 2010.

**Figure 15 polymers-11-00425-f015:**
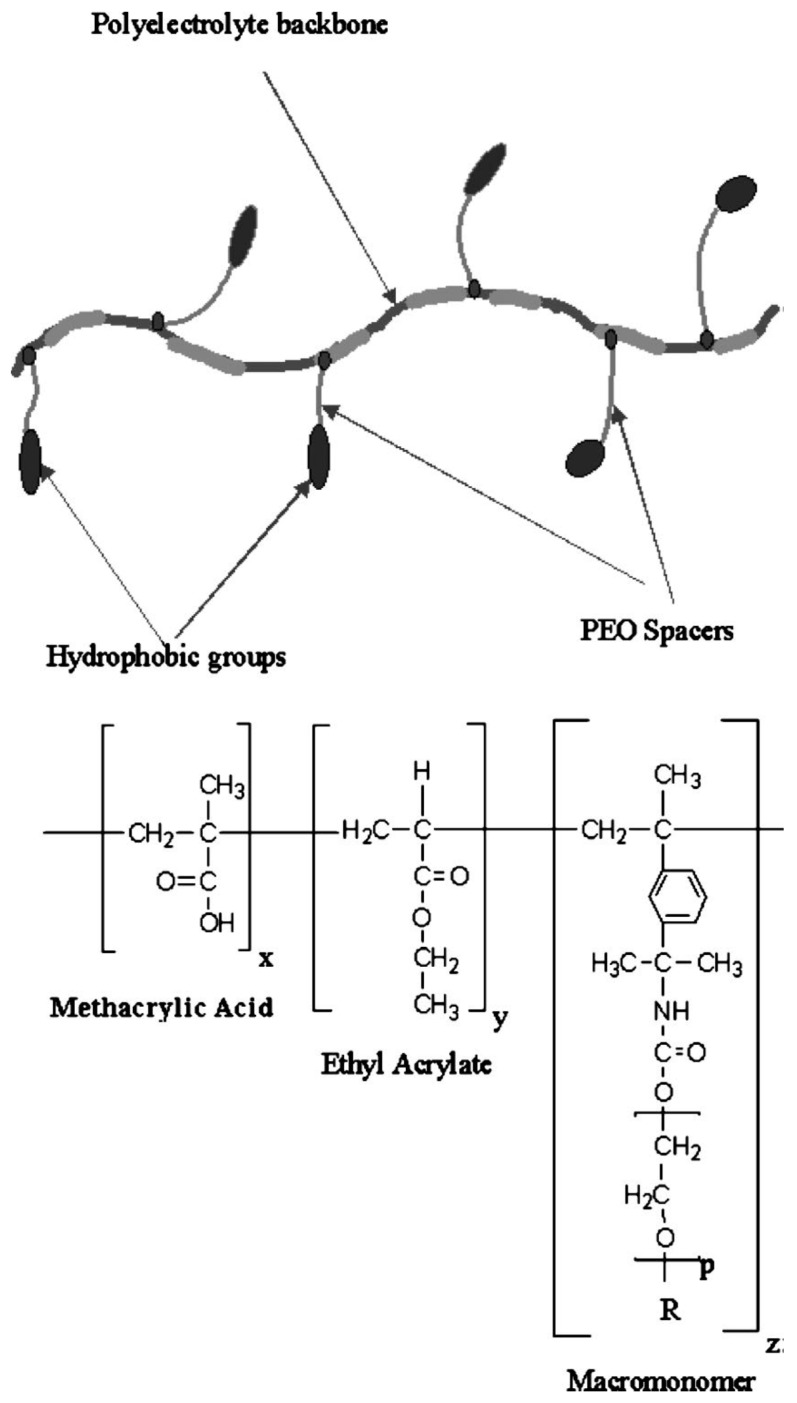
Schematic representation of a typical HASE polymer together with the molecular constitution of the HASE polymers used in this study. *R* refers to the hydrophobic groups. *x*, *y*, *z*, and *p* are structural parameters. Reproduced from [16] with permission, copyright American Chemical Society, 2015.

**Figure 16 polymers-11-00425-f016:**
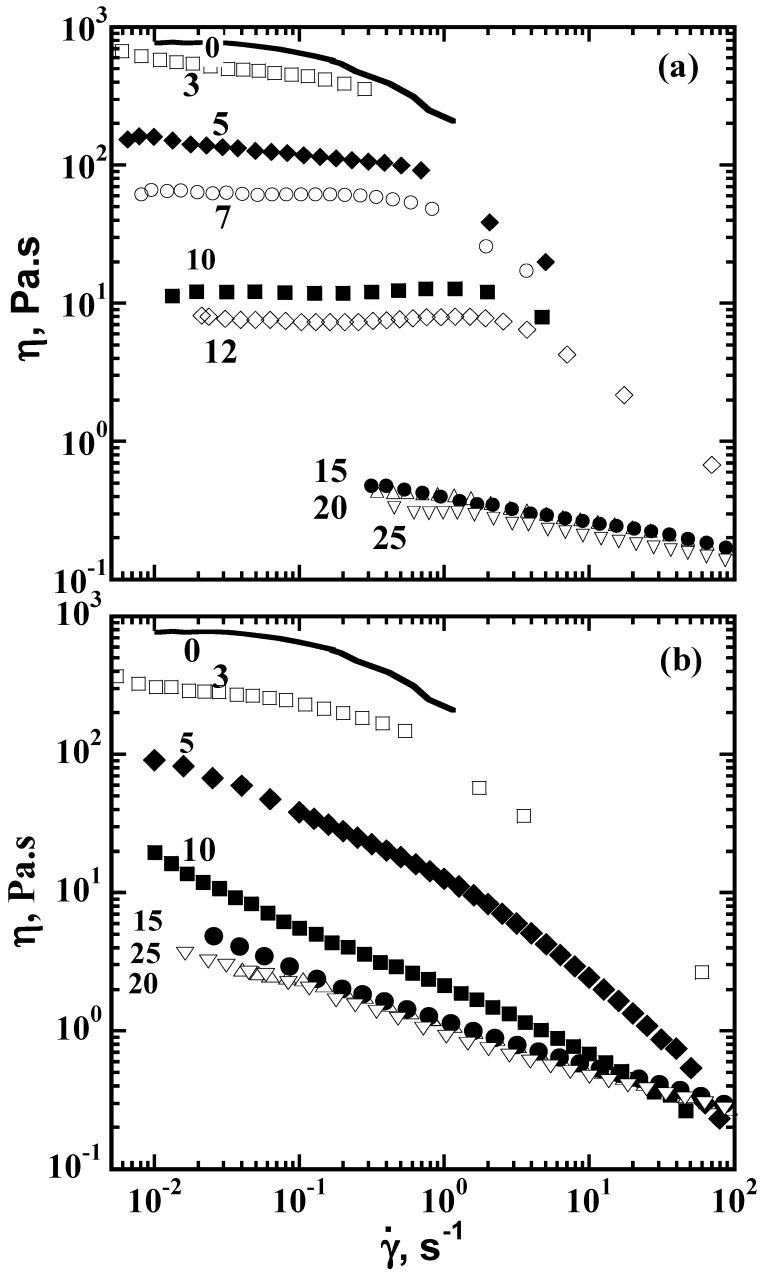
Effects of addition of (**a**) α-CD and (**b**) β-CD on the steady shear viscosity of 3% HASE associative polymer solutions. Numbers correspond to the moles of cyclodextrin per moles of hydrophobe. Reproduced from [15] with permission, copyright John Wiley & Sons, 2013.

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
