# Peer review of "Polymers Containing Non-Covalently Bound Cyclodextrins"

_polymers, 2019, doi:10.3390/polym11030425_

Round 1

Reviewer 1 Report

The manuscript describes the preparation, characterization, properties and potential applications of non-stoichiometric polymer-cyclodextrin-inclusion compounds.

The review focuses on a highly specific topic and it is based on a very limited amount of references. Most of the discussion relates to a single example of polymer containing non-covalently bound cyclodextrins, i.e. nylon 6.

The review could be improved by citing (at least in the introduction part) examples of other polymers that can form inclusion complexes with cyclodextrins.

Author Response

Response to Reviewer 1:

The manuscript describes the preparation, characterization, properties and potential applications of non-stoichiometric polymer-cyclodextrin-inclusion compounds.

The review focuses on a highly specific topic and it is based on a very limited amount of references. Most of the discussion relates to a single example of polymer containing non-covalently bound cyclodextrins, i.e. nylon 6.

The review could be improved by citing (at least in the introduction part) examples of other polymers that can form inclusion complexes with cyclodextrins.

Response: In the introduction I’ve mentioned that many polymers form both stoichiometric- and (n-s)-CD-ICs, and I ve chosen to illustrate them using primarily nylon-6.

Reviewer 2 Report

In the present review, behaviours and smart functions of non stoichiometric polymers included in cyclodextrins are described. The resulting supramolecular polymeric material is based on selective molecular recognition between polymer and cyclodextrin that impart unique functional properties, different from the parent polymer.

The Author, the most expert in the field, analyses, by different spectroscopic and analytical methodologies, the diverse properties of chain portions of polymer when they are included and non-included in cyclodextrin.

Based on non-covalent bonds and on possibility of governing mechanical properties, the supramolecular polymeric material has far greater flexibility allowing interesting industrial applications. Sustainability is another point in favour of the supramolecular system.

The whole structure of the review is reasonable and well organized. It deserves to be published, no doubt.

Some points need more details.

1.    In Figure 7, channel structure of a columnar CD-IC is depicted. Head-to-head and tail-to-tail CD assembly should be mentioned and discussed.

2.    Reference n.7 is missed in the text.

3.    End page is missed in References 1-5 and 11.

4.    Journal name is missed in Reference 18.

Author Response

Response to Reviewer 2:

In the present review, behaviours and smart functions of non stoichiometric polymers included in cyclodextrins are described. The resulting supramolecular polymeric material is based on selective molecular recognition between polymer and cyclodextrin that impart unique functional properties, different from the parent polymer.

The Author, the most expert in the field, analyses, by different spectroscopic and analytical methodologies, the diverse properties of chain portions of polymer when they are included and non-included in cyclodextrin.

Based on non-covalent bonds and on possibility of governing mechanical properties, the supramolecular polymeric material has far greater flexibility allowing interesting industrial applications. Sustainability is another point in favour of the supramolecular system.

The whole structure of the review is reasonable and well organized. It deserves to be published, no doubt.

Some points need more details.

1.    In Figure 7, channel structure of a columnar CD-IC is depicted. Head-to-head and tail-to-tail CD assembly should be mentioned and discussed.

Response: Whether or not the columnar packing of CDs is head-to-head:tail-to-tail or some other stacking is irrelevant to the channel diameter, which I’m discussing using Figure 7.

2.    Reference n.7 is missed in the text.

Response:I’ve now included ref. 7 in text.

3.    End page is missed in References 1-5 and 11.

Response:End pages in References 1-5 and 11 have been added.

4.    Journal name is missed in Reference 18.

Response:I’ve added J. Rheol.

Round 2

Reviewer 1 Report

Although the author has not extensively described other examples of polymers forming inclusion complexes with cyclodextrins, as it was suggested in the first review report, it is also true that references to other papers are correctly listed in the introduction part.

In light of the above and the overall quality of the manuscript, I believe the review is worth of publication and it can be accepted in its present form.